# Efficient nonlinear beam shaping in three-dimensional lithium niobate nonlinear photonic crystals

Dunzhao Wei[1,4], Chaowei Wang[2,4], Xiaoyi Xu[1,4], Huijun Wang[1], Yanlei Hu[2], Pengcheng Chen[1], Jiawen Li[2], Yunzhi Zhu[1], Chen Xin[2], Xiaopeng Hu[1], Yong Zhang [1], Dong Wu[2], Jiaru Chu[2], Shining Zhu[1] & Min Xiao[1,3]

Nonlinear beam shaping refers to spatial reconfiguration of a light beam at a new frequency, which can be achieved by using nonlinear photonic crystals (NPCs). Direct nonlinear beam shaping has been achieved to convert second-harmonic waves into focusing spots, vortex beams, and diffraction-free beams. However, previous nonlinear beam shaping configurations in one-dimensional and two-dimensional (2D) NPCs generally suffer from low efficiency because of unfulfilled phase-matching condition. Here, we present efficient generations of second-harmonic vortex and Hermite-Gaussian beams in the recently-developed three-dimensional (3D) lithium niobate NPCs fabricated by using a femtosecond-laser-engineering technique. Since 3D $\chi^{(2)}$ modulations can be designed to simultaneously fulfill the requirements of nonlinear wave-front shaping and quasi-phase-matching, the conversion efficiency is enhanced up to two orders of magnitude in a tens-of-microns-long 3D NPC in comparison to the 2D case. Efficient nonlinear beam shaping paves a way for its applications in optical communication, super-resolution imaging, high-dimensional entangled source, etc.

[1] National Laboratory of Solid State Microstructures, College of Engineering and Applied Sciences, School of Physics, and Collaborative Innovation Center of Advanced Microstructures, Nanjing University, Nanjing, China. [2] CAS Key Laboratory of Mechanical Behavior and Design of Materials, Key Laboratory of Precision Scientifc Instrumentation of Anhui Higher Education Institutes, Department of Precision Machinery and Precision Instrumentation, University of Science and Technology of China, Hefei, China. [3] Department of Physics, University of Arkansas, Fayetteville, AR, USA. [4] These authors contribute equally: Dunzhao Wei, Chaowei Wang, Xiaoyi Xu. Correspondence and requests for materials should be addressed to Y.Z. (email: zhangyong@nju.edu.cn) or to D.W. (email: dongwu@ustc.edu.cn) or to M.X. (email: mxiao@uark.edu)

Nonlinear photonic crystals (NPCs), featuring spatially-modulated second-order nonlinear coefficients $\chi^{(2)}$, have been widely used to achieve quasi-phase-matching (QPM) for efficient laser frequency conversions[1–4]. Periodically-poled lithium niobite (LiNbO$_3$) crystal is one of the most popular NPCs, which is normally fabricated by using electric poling technique[5,6]. Since the concept of NPC[7] was theoretically extended from one-dimensional to two-dimensional (2D) in 1998, interesting nonlinear optical phenomena, such as cascaded nonlinear optical processes, nonlinear Cerenkov radiation, nonlinear Talbot self-imaging, and quantum path-entanglement, have been extensively investigated in the past two decades[6,8–14].

Along with the development of advanced QPM theories, NPCs with complicated $\chi^{(2)}$ structures have been utilized for nonlinear beam shaping strategies[15–19]. Beam shaping in linear optics has been widely applied to produce various spatial beams for applications in optical communication, optical tweezers, and optical microscopy[20–25]. The nonlinear version, i.e., shaping nonlinear beam at a new frequency, provides a compact way to perform frequency conversion and wavefront manipulation at the same time. However, the frequency conversion efficiency is normally sacrificed for achieving nonlinear beam shaping in previous methods[26–30]. In traditional electric poling technique, the modulations of $\chi^{(2)}$ in NPCs are limited to two dimensions. High conversion efficiency requires one dimension for QPM while nonlinear beam shaping in free space requires two additional dimensions for full wavefront modulations. Therefore, except for few cases that investigate nonlinear beam characteristics within 2D space[16,17,31], most previous works have suffered from low conversion efficiency. The general scheme is to input a fundamental light beam along the $z$-axis of a LiNbO$_3$ NPC and utilize the 2D binary phase modulation of a second-harmonic (SH) wavefront in the $x$–$y$ plane (Fig. 1a).

This configuration has been applied to generate SH vortex beam, nonlinear superfocusing spot, SH non-diffracting beam, etc[27–29,32–34]. Unfortunately, the conversion efficiencies are typically below $1 \times 10^{-6}$ in a mm-long sample due to the lack of phase matching and the use of a small nonlinear coefficient $d_{22}$ of the LiNbO$_3$ crystal, which badly hampers the practical applications of nonlinear beam shaping.

The recent breakthrough in three-dimensional (3D) NPC fabrication techniques[35–37] provides an excellent new platform for achieving efficient nonlinear beam shaping. Such 3D NPCs with full 3D modulations of $\chi^{(2)}$ are capable to simultaneously shape nonlinear beam profiles and fulfill the needed QPM condition at the same time. In this work, we fabricate the specially-designed 3D LiNbO$_3$ NPCs by selectively erasing ferroelectric domains with a femtosecond laser[35,38]. By properly setting the orientation of the 3D NPC structure, the biggest nonlinear coefficient $d_{33}$ of LiNbO$_3$ crystal is utilized under our experimental configuration. In our experiment, the conversion efficiency of nonlinear beam shaping in a tens-of-microns-long 3D NPC is enhanced up to two orders of magnitude higher than the values achieved in the 2D cases.

## Results

**Design of the 3D NPC structure for nonlinear beam shaping.** The 3D NPC provides an extra dimension to achieve full QPM and nonlinear beam shaping simultaneously[39–41]. By using the femtosecond laser erasing technique, one can engineer the amplitude of the nonlinear coefficient $d_{33}$ in a LiNbO$_3$ crystal, which is modulated to $(1-\nu)\,d_{33}$ in the laser illuminated area[35,38]. Here, $\nu$ is defined as the modulation depth. The 3D distribution of $\chi^{(2)}$ in the nonlinear crystal is written as

$$\chi^{(2)}(x,y,z) = d_{33} - d_{33}(1-\nu)f(x,y,z). \quad (1)$$

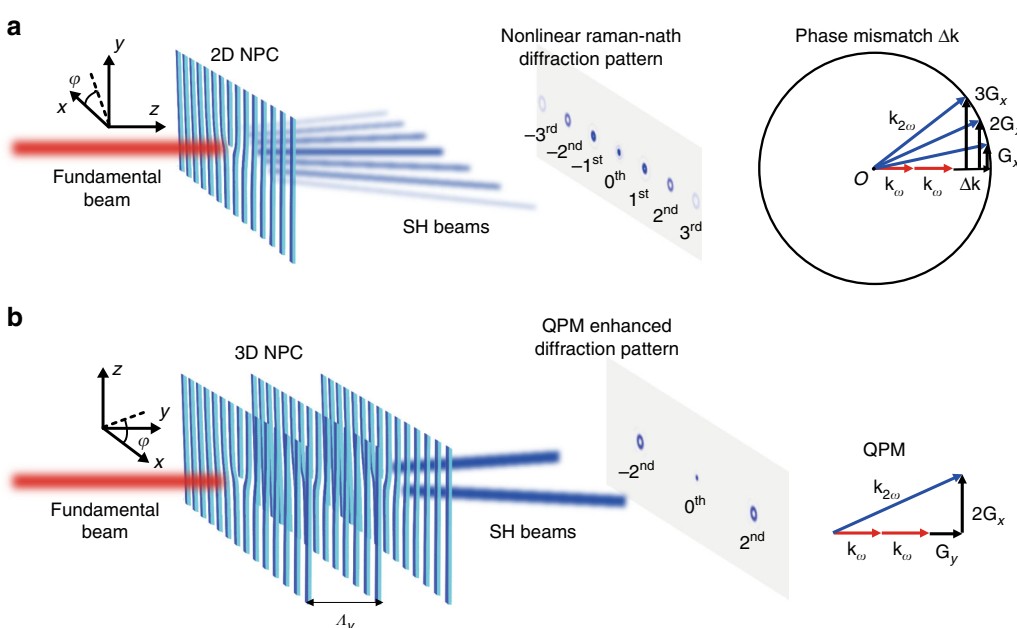

**Fig. 1** Design principle of 3D NPCs. **a** Nonlinear Raman-Nath diffraction orders are generated when a fundamental beam is incident on a 2D nonlinear fork-grating NPC. The SH diffraction pattern exhibits multiple vortex beams with intensity reducing from low to high diffraction orders. However, there exist momentum mismatches **Δk** in the propagation directions for various diffraction orders in nonlinear Raman-Nath diffraction. **b** A 3D nonlinear fork-grating array provides the needed reciprocal vector to compensate momentum mismatch in the propagation direction, so that the selected diffraction orders of SH beams are greatly enhanced. Here, the ±2nd orders are depicted for example. In our experiment, $\Lambda_y = 3\,\mu m$. NPC, nonlinear photonic crystal; SH, second-harmonic; QPM, quasi-phase-matching; 2D, two-dimensional; 3D, three-dimensional

In our configuration, a fundamental wave propagates along the $y$-axis and polarizes along the $z$-axis to utilize the maximal nonlinear coefficient $d_{33}$. Based on a binary computer-generated-hologram theory[42] and assuming a reference plane wave of $e^{iG_xx}$, the structure function $f(x, y, z)$ of a 3D NPC for efficiently generating a SH beam of $E_{2\omega}$ is given by[29] (see Supplementary Note 1 for details)

$$f(x,y,z) = T\left\{\cos[G_xx - \arg(E_{2\omega})] - \cos\left[\sin^{-1}\text{amp}(E_{2\omega})\right]\right\}$$
$$\times T\left[\cos\left(G_yy\right)\right],$$

$$(2)$$

where function $T$ is defined as

$$T(X) = \begin{cases} 1, X \geq 0 \\ 0, X < 0 \end{cases}.$$

$$(3)$$

$G_x$ and $G_y$ are the spatial frequencies along $x$- and $y$-axes, respectively. The "arg" and "amp" functions denote the phase and amplitude of the target SH beam. Equation (2) can be described in terms of Fourier series as

$$f(x,y,z) \propto \sum_{m,n} c_{m,n} \sin\left[m \times \sin^{-1}\text{amp}(E_{2\omega})\right] \times e^{i\left(mG_xx + nG_yy\right)}e^{-im\arg(E_{2\omega})}.$$

$$(4)$$

Here, $m$ and $n$ are integers, and $c_{m,n}$ is the corresponding coefficient. One can deduce from Eq. (4) that the target SH beam exists at the first diffraction order ($m = 1$). In addition, the phase term $e^{i(mG_xx + nG_yy)}$ provides a reciprocal vector $mG_x + nG_y$ to fulfill the QPM condition. It should be noted that the binarization process in structure design also produces a conjugate SH beam at the diffraction order of $m = -1$.

Next, we take SH vortex beams as the target. For comparison, we start from theoretical analysis of a 2D nonlinear beam shaping and then extend it to the 3D case. The traditional electric poling method[27,30] produces a 2D fork-grating domain structure with a topological charge (TC) of $l = 1$ in the $x$–$y$ plane of a LiNbO₃ crystal (Fig. 1a). When a fundamental light beam propagates along the $z$-axis, the generated SH waves experience 0 and $\pi$ phases in the positive and negative domains[43], respectively. In the negative domains, the equivalent modulation depth of $\chi^{(2)}$ is $v = -1$. Therefore, the distribution of relevant nonlinear coefficient in a 2D-poled fork-grating NPC can be written as $d_{22}\{1 - 2T[\cos(G_xx - \varphi)]\} = d_{22}\sum_m c_m e^{imG_xx}e^{-im\varphi}$ (see

Supplementary Note 1 for details). Such binary phase modulation can be used to produce a target SH beam with a vortex phase of $e^{-i\varphi}$ (i.e., $l = 1$) at the first diffraction order. Interestingly, the higher-order nonlinear diffractions from such 2D fork-grating NPC can produce SH vortex beams with higher TCs. The principle can also be understood from the view of nonlinear Raman-Nath diffraction[44]: a transverse reciprocal vector of $mG_x$ contributes to the $m$th diffraction order of SH wave carrying a TC of $l = m$. The corresponding power weight is given by $|c_m|^2$. As shown in Fig. 1a, multiple SH vortex beams are generated at different diffraction orders. However, due to severe phase mismatching along the propagation direction (Fig. 1a), the conversion efficiency is typically below $1 \times 10^{-6}$.

The 3D NPC can provide an additional reciprocal vector along the propagation direction of the fundamental beam to achieve full QPM for nonlinear beam shaping. From Eq. (2), the 3D $\chi^{(2)}$ distribution for SH vortex beam generation can be expressed as

$$\chi^{(2)}(x,y,z) = d_{33} - d_{33}(1 - v)T[\cos(G_xx - \varphi)] \times T\left[\cos\left(G_yy\right)\right]$$
$$= d_{33} - d_{33}(1 - v)\sum_{m,n} c_{m,n} e^{i\left(mG_xx + nG_yy\right)}e^{-im\varphi}.$$

$$(5)$$

It should be noted that due to the use of $d_{33}$, the Cartesian coordinates for our 3D NPC (Fig. 1b) orient differently from those in the 2D case (Fig. 1a). Considering the reciprocal vectors provided by the phase term $e^{i(mG_xx + nG_yy)}$ in Eq. (5), the full QPM condition is given by

$$k_{2\omega} - 2k_\omega - mG_x - nG_y = 0,$$

$$(6)$$

where $k_\omega$ and $k_{2\omega}$ are wavevectors of the fundamental beam and the diffracted SH beam, respectively. For 3D nonlinear beam shaping, the laser-engineered nonlinear fork-grating structure in the $x$–$z$ plane is used to introduce wavefront shaping into the SH beams, which is similar to the 2D case. The advantage of 3D nonlinear beam shaping lies in that the target SH beam is significantly enhanced because the longitudinal phase mismatch can be compensated for the designed diffraction order. Figure 1b shows an example case that the longitudinal reciprocal vector $G_y$ compensates the phase mismatches for the $\pm$2nd nonlinear diffraction orders.

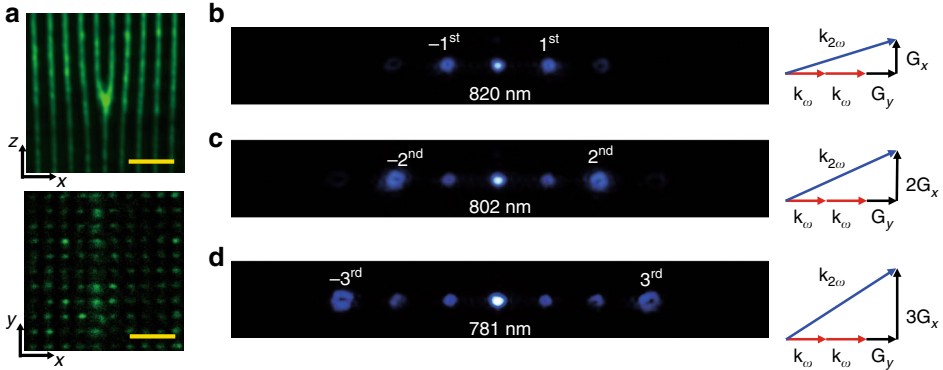

**Fig. 2** Far-field diffraction patterns of the 3D nonlinear fork-grating array. **a** The 3D NPC structures in the $x$–$z$ and $x$–$y$ planes through a confocal SH microscopic system (the length of scale bars, 10 μm). **b**–**d** SH diffraction patterns and their corresponding QPM configurations. The $\pm$1st, $\pm$2nd, and $\pm$3rd diffraction orders are enhanced through noncollinear QPM processes at the input wavelengths of 820 nm, 802 nm, and 781 nm, respectively. In our experiment, $G_x = G_y = 2\pi/3\ \mu m^{-1}$. The 0th-order SH beam is brighter than high-order diffraction beams because of larger interaction volume in a collinear SH generation process

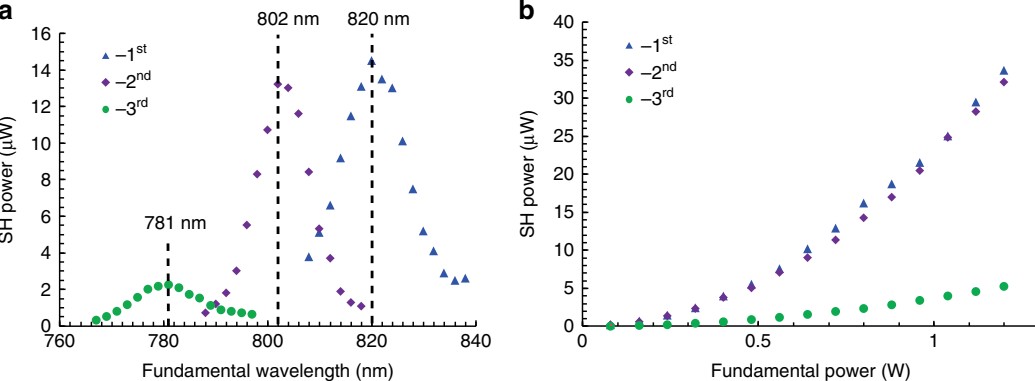

**Fig. 3** Efficient SH vortex beam shaping. **a** Dependence of output powers of the 1st, 2nd, and 3rd diffraction orders on pump wavelength. The pump power is kept at 0.8 W. The peak values indicate the effect of QPM. **b** Dependence of output powers of the 1st, 2nd, and 3rd diffraction orders on pump power at the QPM wavelengths of 820 nm, 802 nm, and 781 nm, respectively

**Efficient SH vortex beam generation**. The 3D fork-grating NPC (Fig. 1b) fabricated by the femtosecond-laser-erasing technique (see "Methods" section for details) has dimensions of 45 μm $(x) \times 45$ μm $(y) \times 45$ μm $(z)$. Figure 2a shows the SH microscopy images[45] of the 3D NPC. The nonlinear structure in the $x$–$z$ plane presents a forkgrating carrying a TC of $l = 1$, while the $x$–$y$ plane shows a 2D periodical array. A fundamental beam generated by a Ti:sappire femtosecond laser (80 MHz repetition rate; 75 fs pulse width) with a tunable wavelength from 690 nm to 1050 nm is used to realize 3D nonlinear beam shaping. It polarizes along the $z$-axis and propagates along the $y$-axis of the crystal. Figure 2b–d show the generated SH diffraction patterns under different input pump wavelengths. The ±1st, ±2nd, and ±3rd diffraction orders of SH vortex beams are phase matched at 820 nm, 802 nm, and 781 nm, corresponding to the vortex beams carrying TCs of $l = ±1$, ±2, and ±3, respectively. See Supplementary Note 3 and Supplementary Fig. 1 for the measurements of TCs. For example, when the input wavelength is 820 nm, the ±1st nonlinear diffraction orders of vortex beams are significantly enhanced with the assistance of the reciprocal vector $±\mathbf{G}_x + \mathbf{G}_y$, distinguishing itself from the traditional nonlinear Raman-Nath diffraction. Similarly, the ±2nd and ±3rd orders of vortex beams (Fig. 2c, d) are phase matched by the reciprocal vectors $±2\mathbf{G}_x + \mathbf{G}_y$ and $±3\mathbf{G}_x + \mathbf{G}_y$, respectively. Obviously, the introduction of the longitudinal reciprocal vector component $\mathbf{G}_y$ in the propagation direction is the key to achieve desired efficient nonlinear beam shaping.

Figure 3a presents the output power dependences of three different nonlinear diffraction orders on the input wavelengths. The input powers are set to be 0.8 W for all the measurements. One can obtain the respective QPM wavelengths from the peaks of the curves in Fig. 3a, which agree well with the theoretically calculated values of 819 nm, 801 nm, and 776 nm, respectively (see Supplementary Note 2 for calculations). A shorter fundamental wavelength is phase matched at a larger emit angle by involving a bigger transverse reciprocal vector. As shown in Fig. 2d, when the ±3rd-diffraction-order SH vortex beams are phase matched, one can still see weak lower-diffraction-order SH beams. This can be attributed to the bandwidth of the fundamental beam (Fig. 3a), which can tolerate certain phase mismatch. Figure 3b shows the dependences of output power on the pump power for different diffraction orders at their respective QPM wavelengths. The quadratic curves agree well with the theoretical relationship between the SH power and the fundamental power. When the

input fundamental power is 1.2 W, the conversion efficiency reaches $2.8 \times 10^{-5}$ for the 1st diffraction order, which is enhanced by one order of magnitude comparing to previous 2D nonlinear beam shaping experiments. The 2nd and 3rd diffraction orders carrying higher topological charges have conversion efficiencies of $2.7 \times 10^{-5}$ and $0.44 \times 10^{-5}$, respectively. The normalized conversion efficiencies are $1.4 \times 10^{-10}$ W$^{-1}$, $1.35 \times 10^{-10}$ W$^{-1}$, and $0.22 \times 10^{-10}$ W$^{-1}$ for the 1st, 2nd and 3rd diffraction order, respectively (see Supplementary Note 4 for calculations of the conversion efficiency). Such efficient SH vortex beam generations can profoundly enrich the applications of NPCs in spatial light beam manipulations and related fields[46–50].

**Efficient Hermite-Gaussian (HG) beam generation in a 3D NPC**. In addition, we design a 3D NPC structure to directly generate SH HG beams in an efficient way. A HG($p$, $q$) mode is given by[51]

$$E_{2\omega}^{p,q}(x, z) = H_p\left(\sqrt{2}\frac{x}{w_0}\right)\exp\left(-\frac{x^2}{w_0^2}\right)H_q\left(\sqrt{2}\frac{z}{w_0}\right)\exp\left(-\frac{z^2}{w_0^2}\right),$$

(7)

where $w_0$ is the beam waist; $p$ and $q$ determine the numbers of nodes in the Hermite polynomials along the $x$-axis and $z$-axis, respectively. We generate a HG(1, 1) as a demonstration, which is expressed as

$$E_{2\omega}^{1,1}(x, z) = \frac{8xz}{w_0^2}\exp\left(-\frac{x^2}{w_0^2}\right)\exp\left(-\frac{z^2}{w_0^2}\right).$$

(8)

According to Eqs (1) and (2), one can obtain a nonlinear structure for generating desired HG beams. Figure 4a depicts the fabricated 3D NPC structure to produce the SH HG(1, 1) mode, presenting four grating structures in the $x$–$z$ plane. The distribution of $\chi^{(2)}$ is mathematically expressed as

$$\chi^{(2)}(x, y, z) = d_{33} - d_{33}(1 - v)$$
$$\times T\{\cos[G_x x - \arg(E_{2\omega}^{1,1})] - \cos[\sin^{-1}\mathrm{amp}(E_{2\omega}^{1,1})]\}$$
$$\times T[\cos(G_y y)].$$

(9)

The "arg" and "amp" functions denote the phase and amplitude of the target SH HG(1, 1) beam. The period along the $x$-direction is 3 μm in each grating. The $x$–$z$ cross-section of the 3D NPC has dimensions of about 32 μm × 32 μm. Along the

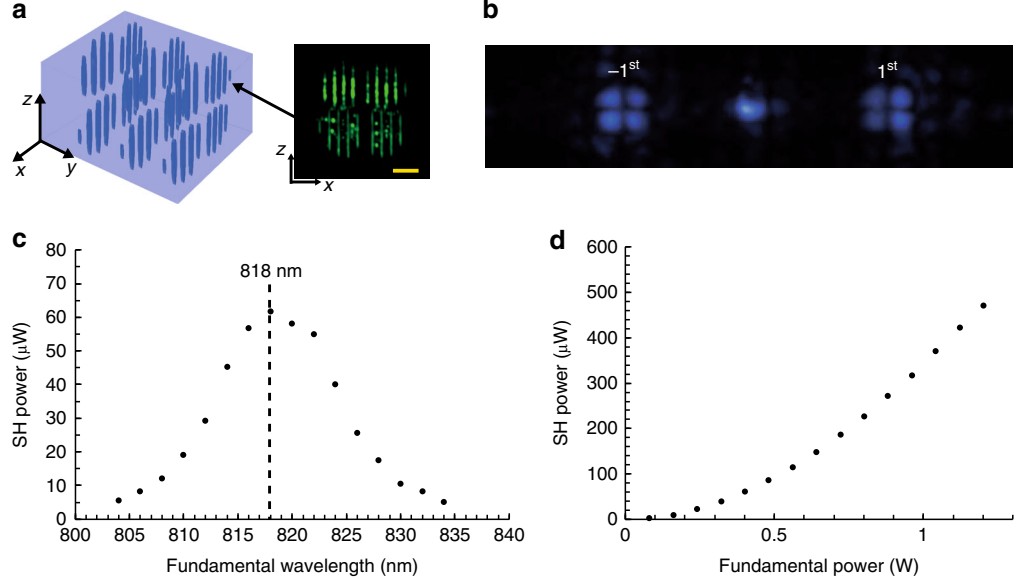

**Fig. 4** Efficient SH HG beam generation. **a** The 3D NPC model and the confocal SH image of its cross-section in the $x$–$z$ plane (the length of scale bar, 10 μm). **b** The SH diffraction pattern pumped under the QPM wavelength of 818 nm. **c** Dependence of output power of the 1st diffraction order on the fundamental wavelength at a pump power of 0.4 W. **d** Dependence of output power of the SH HG beam on pump power at the wavelength of 818 nm. The maximal conversion efficiency reaches to ~$3.9 \times 10^{-4}$ for the 1st diffraction order

$y$-axis, there are 30 periods with an interval of 3 μm. Figure 4b shows the generated SH patterns. The petal-like intensity distributions in the ±1st diffraction orders agree well with the theoretically predicted HG(1, 1) beam. Figure 4c presents the output power dependence of the SH HG beam on the pump wavelength. Clearly, the SH power reaches its peak at 818 nm, which indicates that QPM comes into action. As shown in Fig. 4d, when the pump power is 1.2 W, the conversion efficiency is measured to be ~$3.9 \times 10^{-4}$ (the normalized conversion efficiency is $1.95 \times 10^{-9}$ W$^{-1}$) for each SH HG mode, which is two orders of magnitude higher than the value obtained for the 2D case[29].

## Discussion

We have experimentally demonstrated efficient SH beam shaping in 3D NPCs fabricated by using a femtosecond laser to selectively erase $\chi^{(2)}$ inside a LiNbO$_3$ crystal. Considering the binary-amplitude modulated $\chi^{(2)}$, the 3D NPC structure is theoretically designed according to the computer-generated-hologram method. In the experiment, we have successfully fabricated different 3D nonlinear grating structures to produce the target SH vortex and HG beams. The generated SH beam profiles agree well with the theoretical predictions. The most unique characteristic feature of our 3D NPC is its ability to realize the desired nonlinear beam shaping and full QPM simultaneously. In our experiment, the conversion efficiency of SH beam shaping is enhanced up to two-orders of magnitude in comparison to the previous 2D cases. Considering that our 3D LiNbO$_3$ NPC is only less than 100 μm in length, such enhancement is substantial. In addition, the conversion efficiency can be further improved by upgrading the laser writing system to fabricate a mm-long 3D NPC. Our experimental configuration provides a useful platform to fully utilize the advantages of 3D NPCs, which can be further extended to generate other types of nonlinear beams, nonlinear holographic imaging, and high-dimensionally steered photon entanglements[52–55].

## Methods

**3D NPC fabrication**. We utilize a typical femtosecond-laser writing system for fabricating 3D microstructures in 5% MgO-doped LiNbO$_3$ crystals. The light source of the system is a mode-locked Ti:sapphire laser collocated with a regenerative amplifier (Legend Elite-1K-HE Coherent). This laser produces high intensity pulses at 800 nm wavelength, with a 104 fs pulse duration at 1 kHz pulse repetition rate. In our system, we use a half-wave plate and a Glan Prism to control the power of the laser beam. The laser beam is focused into the LiNbO$_3$ crystal by an objective (50×, N.A. = 0.8) along the $z$-axis after being expanded by a beam expander system. The size of the focal spot inside the crystal is approximately 1.5 μm in X and Y directions and 3 μm in Z direction, which decides the laser writing resolution. During femtosecond processing, the sample position is precisely controlled by a nanopositioning stage (model E-545, Physik Instrumente GmbH & Co. KG) with moving ranges of 200 μm ($x$) × 200 μm ($y$) × 200 μm ($z$) and a spatial resolution of 1 nm. We can observe the fabricating process through a Charge-Coupled-Device camera in real time. The writing energy is 180 nJ with scanning speed 50 μm s$^{-1}$ to fabricate 3D nonlinear fork-gratings for SH vortex beam shaping. To fabricate the 3D NPC structure for SH HG(1, 1) beam generation, the laser writing energies are 110 nJ and 150 nJ for the top and bottom layers along the $z$-axis, respectively. The scanning speed is 50 μm s$^{-1}$. The written volumes are limited by the instability of the femtosecond-laser writing system.

**Experimental setup for characterizing SH beam shaping**. The fundamental (pump) beam is produced by a Ti:sapphire femtosecond laser (Chameleon, Coherent) with a 75 fs pulse duration, 80 MHz pulse repetition rate, and a tunable wavelength ranging from 690 to 1050 nm. Its power is controlled by a half-wave plate and a polarizing beam splitter. The input fundamental beam is injected into the crystal with polarization along the $z$-axis so that the largest nonlinear coefficient $d_{33}$ in the 3D LiNbO$_3$ NPCs is used. The polarization direction is controlled by another half-wave plate. The input beam is focused by a 75 mm lens and incident into the sample. The diameter of the beam waist at the focal point is about 40 μm. Diffraction pattern of the fundamental wave indicates a small change in refractive index (see Supplementary Note 5 and Supplementary Fig. 2 for details). Output far-field SH patterns are projected onto a white screen and then recorded by a camera. A power meter is used to measure the SH power from different diffraction orders under different input wavelengths.

## Data availability

The data that supports the results within this paper and other findings of the study are available from the corresponding authors upon reasonable request.

## Code availability

The custom code and mathematical algorithm used to obtain the results within this paper are available from the corresponding authors upon reasonable request.

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

## Acknowledgements

This work was supported by the National Key R&D Program of China (2017YFA0303703, 2017YFB1104300, 2016YFA0302500, and 2018YFB1105400), the National Natural Science Foundation of China (NSFC) (11874213, 91636106, 11621091, 11674171, 61675190 and 51675503), Youth Innovation Promotion Association CAS (2017495), and the Key R&D Program of Guangdong Province (Grant No. 2018B030329001).

## Author contributions

Y.Z. conceived the idea and organized the project. D.Z.W., C.W.W., X.Y.X., H.J.W., Y.L.H., P.C.C., J.W.L., Y.Z.Z., X.C. and X.P.H. performed the experiments and theoretical calculation under the guidance of Y.Z., D.W., J.R.C., S.N.Z. and M.X. Y.Z. and M.X. supervised the project. All authors contributed to the discussion of experimental results. D.Z.W., Y.Z. and M.X. wrote the paper with contributions from all co-authors.

## Additional information

**Competing interests:** The authors declare no competing interests.

