## [Peer Review File · Nature Communications]

Reviewers' Comments:

Reviewer #1:

Remarks to the Author:

In this paper the authors apply a new technology of three-dimensional modulation of the second order nonlinear coefficient in lithium niobate (reported by them in ref 35) for nonlinear beam shaping applications. In the last decade the concept of nonlinear beam shaping was developed and several different applications were demonstrated, but the performance was limited since it relied on modulation of only two out of the three available axes of the nonlinear crystal.

Now that three dimensional modulation became available, it enabled the authors to improve the efficiency, by using the d_{33} tensor of the nonlinear coefficient, which could not be used earlier, and in addition to obtain full vectorial phase matching. Specifically, the authors demonstrate two off-axis nonlinear holograms that generate vortex beams and Hermite-Gauss₀₁ beam at the second harmonic. Overall, the paper represents an important development in nonlinear optics, and I therefore recommend accepting it for publication in Nature Communications. However, I have some comments and questions that should be addressed:

1. The total written volume seems fairly small. The authors mention 'tens of microns', whereas their stages is capable of 200 micron movement in each direction. This relatively small volume limits the conversion efficiency, which is supposed to be one of the main advantages. What limits this volume and why larger volumes cannot be fabricated?
2. What is the writing resolution in X, Y and Z, and what limits it?
3. The authors mention the average power of the pump and the overall efficiency, but the important parameters that should be reported are peak power, beam size and normalized conversion efficiency per (peak) pump power. It is also important to compare the measured efficiency with calculated efficiencies in order to estimate the differences between design and fabrication.
4. With respect to generation of vortex beams, earlier publication on on-axis generation of vortex beam should be cited: Bahabad and Arie, Optics Express 15, 17619 (2007).

Reviewer #2:

Remarks to the Author:

The authors demonstrate experimentally nonlinear beam shaping by quadratic nonlinear crystals with 3D engineered domain structures. They use a femtosecond laser writing technique, that was recently developed by them, to erase the quadratic coefficient in specific areas and create the 3D structures in the materials. They key important point of the paper is that by using 3D modulation, the second harmonic beam shaping process can be more efficient than previously demonstrated nonlinear beam shaping by 2D modulated crystals. This is one of the first demonstrations of applications of recently developed 3D nonlinear photonic crystals. Therefore, I believe that it will have a very strong impact and recommend to publish it in Nature Communications journal after some required corrections detailed below.

Required revisions:

The main point of the paper is the better conversion efficiency that can be achieved with respect to the case of nonlinear beam shaping by 2D modulations of the quadratic coefficient. The paper presents some numbers and some comparisons to previous works, however as this is the main novelty of the paper the authors should add either to the main text or maybe to the supplementary materials a part with equations that will show the readers specifically how to calculate the conversion efficiency in each case.

The actual values or upper limit χ^2 in the erased domains should be stated and taken into account.

What was the degree of modification to the refractive index due to the fabrication method. Is it improved with respect to the authors' previous work, and does it affect the conversion efficiency or limit the interaction length.

Showing diffraction images of the FH would be good.

Peak power of the laser and intensity in the crystal should be taken into account in the implicit calculation. Using the theoretical conversion efficiency equations, specific comparison between the experiment and the theory should be made. In addition specific comparison between the efficiency obtained in the case of 2D and 3D modulations should be made. This part should also allow the reader to understand why you claim from 10nm QPM bandwidths.

Clarify why the zero order SH in Fig. 2 is always brighter with respect to the diffracted beams that are phase matched.

It might be good to write explicitly the expression for the HG22. Also say what is arg and amp in eq. 8

Response to the reviewer comments for manuscript

NCOMMS-19-18413:

We are glad to see that both reviewers recommend our manuscript for publication in *Nature Communications*. We sincerely thank the reviewers for their constructive suggestions to further improve the manuscript. In the following, we respond point-by-point to all the comments and questions raised by the reviewers and indicate corresponding changes made in the revised manuscript.

Response to the comments by Reviewer 1:

Comments: In this paper the authors apply a new technology of three-dimensional modulation of the second order nonlinear coefficient in lithium niobate (reported by them in ref 35) for nonlinear beam shaping applications. In the last decade the concept of nonlinear beam shaping was developed and several different applications were demonstrated, but the performance was limited since it relied on modulation of only two out of the three available axes of the nonlinear crystal.

Now that three dimensional modulation became available, it enabled the authors to improve the efficiency, by using the d_{33} tensor of the nonlinear coefficient, which could not be used earlier, and in addition to obtain full vectorial phase matching. Specifically, the authors demonstrate two off-axis nonlinear holograms that generate vortex beams and Hermite-Gauss₀₁ beam at the second harmonic. Overall, the paper represents an important development in nonlinear optics, and I therefore recommend accepting it for publication in *Nature Communications*. However, I have some comments and questions that should be addressed:

Our response: We sincerely appreciate the reviewer's positive comments about our work. In the following, we clarify/address the reviewer's concerns and indicate the revisions made in the revised manuscript and Supplementary Information (SI) accordingly.

Comment 1: The total written volume seems fairly small. The authors mention 'tens of microns', whereas their stages is capable of 200 micron movement in each direction. This relatively small volume limits the conversion efficiency, which is supposed to be one of the main advantages. What limits this volume and why larger volumes cannot be fabricated?

Our response:

In our experiment, the written volumes are limited by the instability of the femtosecond (fs) laser writing system currently used. In comparison to our previous work (i.e., 3D periodic NPC structure), the fabrications of the complicated structures in this work require higher precision and more time in laser writing. However, because of the instabilities of translation stage and laser pulse energy due to the aging of the equipment, our laser writing system cannot afford high-precision fabrication for more than one hour, which severely limits the written volume to be within tens of microns in our current experiment.

It should be noted that our experimental results have demonstrated an enhanced conversion efficiency of nonlinear beam shaping in 3D NPC in comparison to that in 2D NPC. We hope to achieve higher efficiency in larger 3D NPCs by upgrading the laser writing system with a multiple-focus fabrication setup, a highly stable stage, and so on.

In the revised manuscript, we have added “the written volumes are limited by the instability of the femtosecond-laser writing system” in the **Methods: 3D NPC fabrication** to clearly address this issue.

Comment 2: What is the writing resolution in X, Y and Z, and what limits it?

Our response:

Thanks for the useful advice. In our processing system, the beam is focused into lithium niobate crystal through an objective lens. The numerical aperture of the objective lens determines the width of the focal volume and therefore the writing resolution.

After passing through the objective lens, the diameter of focal spot can be calculated as $d=1.22\lambda/NA$, where λ is the wavelength of incident light in fabrication; NA is the numerical aperture of the objective lens. In our experiment, λ is 800 nm and NA is 0.8. Then, the diameter of focal spot in X, Y direction is about 1.2 μm in theory. Due to the slight thermal diffusion effect of femtosecond laser interacting with transparent materials, the writing resolution will exceed the focal diameter. Finally, the writing resolution in X, Y direction is about 1.5 μm . Due to the influence of refractive index mismatch, the writing resolution in Z direction decreases with the increase of processing depth. Within the processing depth of this experiment, the writing resolution in Z direction is about 3 μm .

We have added this information in the **Methods: 3D NPC fabrication**.

Comment 3: The authors mention the average power of the pump and the overall efficiency, but the important parameters that should be reported are peak power, beam size and normalized conversion efficiency per (peak) pump power. It is also important to compare the measured efficiency with calculated efficiencies in order to estimate the differences between design and fabrication.

Our response:

We thank the reviewer for this constructive suggestion. In our experiment, the peak

power of the pump beam is 2×10^5 W and the beam size is 40 μm in diameter. The normalized conversion efficiencies are measured to be 1.4×10^{-10} W^{-1} for the SH OAM beam of $l = 1$ and 1.95×10^{-9} W^{-1} for the SH HG(1,1) beam, which are consistent with the theoretical values of 3.1×10^{-10} W^{-1} and 2.7×10^{-9} W^{-1} , respectively. The differences between the experimental and theoretical results can be attributed to the imperfection in the fabricated 3D NPC structure, as well as the diffraction and scattering losses. The detailed calculations are added in **Supplementary Note 4**.

Comment 4: *With respect to generation of vortex beams, earlier publication on on-axis generation of vortex beam should be cited: Bahabad and Arie, Optics Express 15, 17619 (2007).*

Our response: Taking the reviewer's advice, we have added this paper as the new Ref. 41 in the revised manuscript.

Response to the comments by Reviewer 2:

Comments: *The authors demonstrate experimentally nonlinear beam shaping by quadratic nonlinear crystals with 3D engineered domain structures. They use a femtosecond laser writing technique, that was recently developed by them, to erase the quadratic coefficient in specific areas and create the 3D structures in the materials. The key important point of the paper is that by using 3D modulation, the second harmonic beam shaping process can be more efficient than previously demonstrated nonlinear beam shaping by 2D modulated crystals. This is one of the first demonstrations of applications of recently developed 3D nonlinear photonic crystals. Therefore, I believe that it will have a very strong impact and recommend to publish it in Nature Communications journal after some required corrections detailed below.*

Our response: We sincerely thank the reviewer's positive comments and useful suggestions. In the following, we further clarify the reviewer's concerns and indicate revisions made in the revised manuscript and Supplementary Information (SI) accordingly.

Comment 1: *The main point of the paper is the better conversion efficiency that can be achieved with respect to the case of nonlinear beam shaping by 2D modulations of the quadratic coefficient. The paper presents some numbers and some comparisons to previous works, however as this is the main novelty of the paper the authors should add either to the main text or maybe to the supplementary materials a part with equations that will show the readers specifically how to calculate the conversion efficiency in each case.*

Our response:

We thank for the reviewer's constructive suggestion. In nonlinear beam shaping, the normalized conversion efficiency per peak power can be expressed as

$$\eta_{\text{nor}} = \frac{2d_{\text{eff}}^2 L^2 \omega^2}{n_{\omega}^2 n_{2\omega} \epsilon_0 c^3 A_{\text{eff}}} \text{sinc}^2\left(\frac{\Delta k L}{2}\right). \quad (\text{R1})$$

Here, Δk is the phase mismatch along the propagation direction. Generally, $\Delta k \neq 0$ in 2D nonlinear beam shaping. In 3D case, $\Delta k = 0$ when the QPM condition is satisfied. Therefore, the conversion efficiency can be enhanced in 3D NPC. See **revised Supplementary Note 4** for detailed deduction of Eq. R1.

Comment 2: *The actual values or upper limit chi 2 in the erased domains should be stated and taken into account.*

Our response:

The $\chi^{(2)}$ in the erased domains of our sample is reduced to be about 85% of that in the non-erased ones, which is calculated from the measured SH conversion efficiency. We have added this information in **Supplementary Note 4** and taken it into account in the theoretical calculations.

Comment 3: *What was the degree of modification to the refractive index due to the fabrication method. Is it improved with respect to the authors' previous work, and does it affect the conversion efficiency or limit the interaction length.*

Our response:

The modification of refractive index was measured to be about 5×10^{-3} , which is similar to our previous work. Considering that the refractive index changes for the fundamental and second-harmonic waves in the erased area are nearly the same (see the SI in our previous work), such modification has no significant influence on the interaction length and the QPM condition. Because of the scattering and diffraction induced by the refractive index change, the conversion efficiency is decreased by a few percent in our experiment.

We have added above information in the revised main text and **Supplementary Note 5**.

Comment 4: *Showing diffraction images of the FH would be good.*

Our responds:

Taking the reviewer's suggestion, we have added the diffraction images of the

fundamental wave (FW) in **Supplementary note 5**.

Fig. R2 Diffraction images of the fundamental wave through the 3D NPCs for OAM mode (up) and HG(1,1) mode (down).

Comment 5: *Peak power of the laser and intensity in the crystal should be taken into account in the implicit calculation. Using the theoretical conversion efficiency equations, specific comparison between the experiment and the theory should be made. In addition specific comparison between the efficiency obtained in the case of 2D and 3D modulations should be made. This part should also allow the reader to understand why you claim from 10nm QPM bandwidths.*

Our response: We appreciate the reviewer's helpful suggestion. The peak power (2×10^5 W) and intensity (1×10^3 W·mm⁻²) of the input laser have been taken into account in our calculations. When QPM condition is satisfied, the theoretical normalized conversion efficiency of 3D nonlinear beam shaping can be calculated from

$$\eta_{\text{nor}_3\text{D}} = \frac{2d_{\text{eff}_3\text{D}}^2 L^2 \omega^2}{n_{\omega}^2 n_{2\omega} \epsilon_0 c^3 A_{\text{eff}}} \quad (\text{R2})$$

See revised **Supplementary Note 4** for details of the theoretical calculations.

The normalized conversion efficiencies are measured to be 1.4×10^{-10} W⁻¹ for the SH OAM beam of $l = 1$ and 1.95×10^{-9} W⁻¹ for the SH HG(1,1) beam, which are consistent with the theoretical values of 3.1×10^{-10} W⁻¹ and 2.7×10^{-9} W⁻¹, respectively. The differences between the experimental and theoretical results can be attributed to the imperfections in the 3D NPC structures. The typical conversion efficiency in 2D NPC is 1×10^{-13} W⁻¹ [see Ref. R1 for example]. Consider that the diameters of the fundamental beams are ~ 40 μm in our work and ~ 300 μm in Ref. R1. Under the same pump condition, the conversion efficiency of our 3D nonlinear beam shaping is enhanced by at least one order of magnitude in comparison to the 2D case. In addition, the conversion efficiency of 3D nonlinear beam shaping can be further enhanced by increasing the sample length along the propagation direction, which cannot be achieved in the 2D case. We have added this discussion in Page 7 and 8 of the revised

Manuscript, as well as the revised **Supplementary Note 4**

We apologize for the unclear description about 10 nm QPM bandwidth. In our experiment, the QPM bandwidth mainly results from the bandwidth of the 75 fs laser pulse for SHG process. We have revised the statement to “This can be attributed to the bandwidth of the fundamental beam (Fig. 3a), which can tolerate certain phase mismatch” for clarification.

Comment 6: Clarify why the zero order SH in Fig. 2 is always brighter with respect to the diffracted beams that are phase matched.

Our response:

The reasons can be summarized in two aspects. First, the interaction volume of the zero-order SHG is much larger than the higher-order SHG because (1) the zero-order SH beam is generated by collinear SHG while the higher-order SH beams result from non-collinear SHG, and (2) the zero-order SHG is a Gaussian-FW to Gaussian-SH conversion while the higher-order SHG processes correspond to Gaussian-FW to OAM-SH conversion. Second, the bandwidth of the 75 fs pump laser is about 15 nm. When the fundamental wavelength is tuned away from the QPM wavelength of the zero-order SHG, the phase mismatch is not enough to significantly decrease the power of the zero-order SH beam.

We have clarified this point in Fig. 2 legend and page 7 of the revised **Manuscript**

Comment 7: It might be good to write explicitly the expression for the HG₂₂. Also say what is arg and amp in eq. 8.

Our response:

We apologize for that we made a mistake in the original manuscript. The target HG mode is a SH HG(1, 1) mode instead of SH HG(2, 2) mode in our experiment. We have corrected this in the revised manuscript.

The expression for HG(1, 1) mode is

$$E_{2\omega}^{1,1}(x, z) = \frac{8xz}{w_0^2} \exp\left(-\frac{x^2}{w_0^2}\right) \exp\left(-\frac{z^2}{w_0^2}\right),$$

where w_0 is the beam waist. Following the reviewer’s suggestion, we have added it as Eq. (8) in the revised manuscript (Page 8). The “arg” and “amp” refer to the phase and amplitude of a field, which are defined in the revised manuscript (Page 8).

Reference

- R1 Shapira, A., Shiloh, R., Juwiler, I. & Arie, A. Two-dimensional nonlinear beam shaping. *Opt. Lett.* 37, 2136-2138 (2012).

Change list

1. In page 3, we have added a sentence “NPC, nonlinear photonic crystal; SH, second-harmonic; QPM, quasi-phase matching” in Fig. 1’s legend to define the all abbreviations in Fig. 1.
2. In page 4, we have cited a new Ref. 41, “*Bahabad and Arie, Optics Express 15, 17619 (2007)*”, in the first sentence.
3. In the beginning of Page 6, the subheading “Efficient SH vortex beam generation through 3D nonlinear beam shaping” is shorten to “Efficient SH vortex beam generation” to satisfy the format requirement.
4. In page 7, the unit “ μm^{-1} ” in Fig. 2 legend is revised as “ μm^{-1} ”
5. In page 7, Fig. 2’s legend, we have added a sentence “The 0th-order SH beam is brighter than high-order diffraction beams because of larger interaction volume in a collinear SH generation process” in the end.
6. In page 7, the sentences “This can be attributed to the 10 nm QPM bandwidth (Fig. 3a) in our 3D NPC sample, which can tolerate certain phase mismatch” has been revised as “This can be attributed to the bandwidth of the fundamental beam (Fig. 3a), which can tolerate certain phase mismatch”.
7. In page 7, we have added a sentence “The normalized conversion efficiencies are $1.4 \times 10^{-10} \text{ W}^{-1}$, $1.35 \times 10^{-10} \text{ W}^{-1}$, and $0.22 \times 10^{-10} \text{ W}^{-1}$ for the 1st, 2nd and 3rd diffraction order, respectively”.
8. In page 8, the sentence “A HG(p , q) beam is given by” has been revised as “A HG(p , q) mode is given by”.
9. In page 8, we have added a sentence “We generate a HG(1, 1) as a demonstration, which is expressed as

$$E_{2\omega}^{1,1}(x, z) = \frac{8xz}{w_0^2} \exp\left(-\frac{x^2}{w_0^2}\right) \exp\left(-\frac{z^2}{w_0^2}\right).$$

10. In page 8 and 10, we have revised “HG(2, 2)” to “HG(1, 1)”.
11. In page 8, we have added a sentence “The “arg” and “amp” functions denote the phase and amplitude of the target SH HG(1, 1) beam” to define “arg” and “amp”.
12. In page 8, the last sentence “As shown in Fig. 4d, when the pump power is 1.2 W, the conversion efficiency is measured to be $\sim 3.9 \times 10^{-4}$ for each SH HG mode, which is two orders of magnitude higher than the value obtained for the 2D case²⁹,” has been rewritten as “As shown in Fig. 4d, when the pump power is 1.2 W, the conversion efficiency is measured to be $\sim 3.9 \times 10^{-4}$ (the normalized conversion efficiency is $1.95 \times 10^{-9} \text{ W}^{-1}$) for each SH HG mode, which is two orders of magnitude higher than the value obtained for the 2D case²⁹”.
13. In page 10, **3D NPC fabrication in Methods**, the sentence “The size of the focal spot inside the crystal is approximately 1.2 μm in the transverse direction” in the middle has been rewritten as “The size of the focal spot inside the crystal is approximately 1.5 μm in X and Y directions and 3 μm in Z direction, which

decides the laser writing resolution”.

14. In page 10, **3D NPC fabrication** in **Methods**, the unit “ $\mu\text{m}\cdot\text{s}^{-1}$ ” has been revised as “ $\mu\text{m}\cdot\text{s}^{-1}$ ”.
15. In the reference, the reference “*Bahabad and Arie, Optics Express 15, 17619 (2007).*” numbered as 41 has been added.
16. We add two more sections in Supplementary information, i.e., Supplementary note 4 | Calculation of conversion efficiency and Supplementary note 5 | Diffraction pattern of the fundamental wave.

All the changes are marked in red in the revised manuscript.

Reviewers' Comments:

Reviewer #1:

Remarks to the Author:

The authors have made all the required modifications. The paper can be accepted for publication.

Reviewer #2:

Remarks to the Author:

The authors addressed all my previous comments properly and I have no further comments.

Reviewers' comments

Reviewer #1 (Remarks to the Author):

The authors have made all the required modifications. The paper can be accepted for publication.

Reviewer #2 (Remarks to the Author):

The authors addressed all my previous comments properly and I have no further comments.

Our response:

We appreciate the positive comments of both reviewers.